# Influence of Meteorological Factors on the COVID-19 Transmission with Season and Geographic Location

**DOI:** 10.3390/ijerph18020484

**Published:** 2021-01-09

**Authors:** Xiao-Dong Yang, Hong-Li Li, Yue-E Cao

**Affiliations:** 1Department of Geography and Spatial Information Techniques/Center for Land and Marine Spatial Utilization and Governance Research, Ningbo University, Ningbo 315211, China; yangxiaodong@nbu.edu.cn (X.-D.Y.); nbulihongli@126.com (H.-L.L.); 2Ningbo Universities Collaborative Innovation Center for Land and Marine Spatial Utilization and Governance Research at Ningbo University, Ningbo 315211, China; 3Institute of East China Sea, Ningbo University, Ningbo 315211, China; 4School of Environmental and Geographical Science, Shanghai Normal University, Shanghai 200234, China

**Keywords:** community-based pandemic prevention and control, geographical location, precipitation, relative humidity, season, temperature, wind speed

## Abstract

The purpose of this study is to investigate whether the relationship between meteorological factors (i.e., daily maximum temperature, minimum temperature, average temperature, temperature range, relative humidity, average wind speed and total precipitation) and COVID-19 transmission is affected by season and geographical location during the period of community-based pandemic prevention and control. COVID-19 infected case records and meteorological data in four cities (Wuhan, Beijing, Urumqi and Dalian) in China were collected. Then, the best-fitting model of COVID-19 infected cases was selected from four statistic models (Gaussian, logistic, lognormal distribution and allometric models), and the relationship between meteorological factors and COVID-19 infected cases was analyzed using multiple stepwise regression and Pearson correlation. The results showed that the lognormal distribution model was well adapted to describing the change of COVID-19 infected cases compared with other models (R^2^ > 0.78; *p*-values < 0.001). Under the condition of implementing community-based pandemic prevention and control, relationship between COVID-19 infected cases and meteorological factors differed among the four cities. Temperature and relative humidity were mainly the driving factors on COVID-19 transmission, but their relations obviously varied with season and geographical location. In summer, the increase in relative humidity and the decrease in maximum temperature facilitate COVID-19 transmission in arid inland cities, while at this point the decrease in relative humidity is good for the spread of COVID-19 in coastal cities. For the humid cities, the reduction of relative humidity and the lowest temperature in the winter promote COVID-19 transmission.

## 1. Introduction

The 2019 coronavirus disease (COVID-19) is ascribed to the severe acute respiratory syndrome coronavirus 2 (SARS-CoV-2), a novel coronavirus. Due to high contagiousness, COVID-19 quickly appeared around the world after the first report in Wuhan of Hubei province, China, in late December 2019. The World Health Organization (WHO) announced COVID-19 as a global public health emergency and one of the major human disasters in the 21st century. Through 11 November, 2020, a total of 59,481,313 confirmed cases and 1,404,542 deaths had been reported around the world [1]. It is estimated that the global impact of COVID-19 may last several years [1]. At present, as temperatures fall and winter arrives, the COVID-19 pandemic in many parts of the Northern Hemisphere, such as France, UK, Spain, Italy and Belgium, have seen a second outbreak [2]. For example, since October, more than 10,000 new positive cases have been confirmed every day in many countries [1]. France and Germany announced a second national lockdown policy on October 28, and parts of Spain and Greece have also implemented pandemic prevention and control [2]. Governments and medical and public health systems in many countries would once again face severe challenges of epidemic prevention.

Meteorological factors affect transmission of COVID-19 [3,4,5,6], but their influence varies greatly among different regions [7,8,9,10]. For example, temperature had a significant positive correlation with COVID-19 transmission in Singapore, Brazil, Indonesia, Japan and Norway [5,11,12,13,14], but significantly negatively correlated in New York City, Iran, Bangladesh and China [6,8,15,16], and not significantly related in the United States, Spain and Mexico [17,18,19]. Relative air humidity was positively correlated with COVID-19 transmission in Singapore and Brazil [11,13], but not directly linked in New York and India [6,9]. Precipitation was most conducive to COVID-19 transmission in Norway [5], whereas not significantly affected by the increase in recorded COVID-19 confirmed cases in Brazil, Mexico and many other countries [11,19]. Wind speeds were not promoted the spread of COVID-19 in Singapore and Iran [8,13]; however, high wind speeds accelerated the increase of COVID-19 confirmed cases in Turkey [20]. These studies have shown that the relationship between COVID-19 transmission and meteorological factors varied among different regions [10,21,22]. However, current research has not clearly revealed reasons for this difference.

The difference in relationship between COVID-19 transmission and meteorological factors may be caused by the changes of season and geographical location [23,24]. For example, in arid regions, an increase in relative humidity might be beneficial for virus survival and spread because it reduces drought stress [25]. Oppositely, it might play a negative influence on COVID-19 transmission as it promotes viruses and droplets to combine into larger drops in humid regions [11,25]. The deposition of massive syncretic droplets will reduce virus abundance in the atmosphere [26,27]. In winter, a slight reducing temperature can prolong survival time of virus in atmosphere, thus facilitating virus transmission [18,28,29]. However, as high temperatures inactivates virus via denaturing the capsid protein and the glycoprotein spike, the increase in maximum temperature would prevent virus attaching to the host cells in summer [24,30].

Difference in relationship between COVID-19 transmission and meteorological factors may also be related to government’s epidemic prevention and control measures [31,32]. The effective implementation of epidemic prevention and control might reduce the impact of meteorological factors on COVID-19 transmission, or even lead to no obvious relationship between them [10,33]. At present, a wide variety of measures have been implemented to prevent and control COVID-19 in different regions [1]. For example, China and Italy imposed community lockdown and home quarantine in COVID-19 outbreak cities. The U.S. states of Oregon, Washington, California and New York limited family gatherings and closed public places (such as bars, restaurants and gyms) after 22:00 to control COVID-19. Germany used curfew, the closure of public places, and forbid celebrations in public and private places to prevent COVID-19 during the outbreak period. However, there is no uniform standard of pandemic prevention and control around the world. It is difficult to accurately reveal the influences of season and geographic location on relationship between COVID-19 transmission and meteorological factors.

After December 2019, successive outbreaks of COVID-19 occurred in Wuhan, Harbin, Beijing, Urumqi, Dalian, Qingdao and Kashgar, China. China’s government implemented a series of community-based prevention and control measures, such as traffic control, home quarantine, and nucleic acid testing of all residents and registering digital QR code, to minimize the danger of the virus to local residents [34]. Since pandemic prevention and control is carried out under the guidance of the National Health Commission of China, all COVID-19 outbreak cities adopted the same community management measures. According to the China government report and the published studies, the implementation of these measures effectively limited the spread of COVID-19 [18,32,34,35]. Until now, *R*_0_, the basic reproductive rate indicating the transmissibility of novel coronavirus, is below 1 [36,37]. The death toll is less than 5000 [34]. Because the same community-based pandemic prevention and control measures were adopted in different cities by the Chinese government, the change of relationship between meteorological factors and COVID-19 transmission may be mainly affected by season and geographical location. However, at present there is no study analyzing and testing this speculation.

The aim of this study is to reveal if the influence of meteorological factors on COVID-19 transmission varies with season and geographical location. In order to achieve this scientific aim, we collected the total number of COVID-19 infected people (including the confirmed cases and asymptomatic-infected persons) and meteorological factors in all COVID-19 outbreak cities of China during the period of community-based pandemic prevention and control. Then, we used four statistical models to fit the change of COVID-19 infected cases, and used multiple stepwise regression and Pearson correlation to analyze the relationship between meteorological factors and COVID-19 infected cases. Our study provides a new view for explaining the reasons for COVID-19 transmission while under the implementation of community-based pandemic prevention and control, which is conducive to predicting COVID-19 transmission dynamics, and developing effective prevention and control measures.

## 2. Data Collection

After December 2019, the Chinese government officially reported COVID-19 outbreaks in seven cities (Wuhan, Harbin, Beijing, Urumqi, Dalian, Qingdao and Kashgar) (Table 1). Among these cities, the number of infected cases reported in Harbin and Qingdao was small, thus we did not include these two cities in the data processing due to low statistical significance. At present (11 November 2020), the epidemic has not ended in Kashgar, so we did not study it. Wuhan, Beijing, Urumqi and Dalian were selected as research objects. The epidemic data of each city, including the number of newly confirmed cases and the number of the asymptomatic infections person, were obtained from the China National Health Commission (NHC) (http://www.nhc.gov.cn). Meteorological data were collected from the China National Meteorological Information Center (http://data.cma.cn), which included daily maximum temperature (T_max_), minimum temperature (T_min_), temperature range (DTR), average temperature (T_m_), relative humidity (RH), average wind speed (MWS) and precipitation (TP) (Figure 1).

The aim of this study is to reveal the reason for the change in relationship between COVID-19 transmission and meteorological factors among the four cities after the implementation of community-based epidemic prevention and control. Therefore, the starting time of our study is when epidemic prevention control measures were first imposed, while the end time is when the number of newly confirmed cases returns to zero (Table 1 and Figure 1). Except for the confirmed cases, COVID-19 patients also include those who are asymptomatically infected, which has no clinically identifiable symptoms such as fever, cough and sore throat. Asymptomatically infected persons are difficult to find within the population, but they can spread novel coronavirus to others, or become a confirmed case in the days ahead [2]. Starting on 31 March 2020, the health commissions of Chinese provinces and municipalities began to release data on the asymptomatically infected persons. Therefore, the sum of the confirmed cases and the number of asymptomatically infected persons is defined as the infected cases. However, the occurrence period of the COVID-19 outbreak in Wuhan was earlier than 31 March 2020, so there are no data on asymptomatic infections in Wuhan (Table 1).

## 3. Method

In this study, four statistic models (i.e., Gaussian, logistic, lognormal distribution and allometric models) were first performed on the change of the infected cases. R^2^ and *p*-value were used to select the best-fitting model. The model with larger R^2^ and the smallest *p*-value was the best-fitting model. The difference in the fitting parameters was used to determine if COVID-19 transmission patterns varied among the four cities. Since the best-fitting model was presented as an identical equation, different fitting parameters indicated if COVID-19 transmission patterns varied among the four cities. After that, one-way ANOVA was used to compare the differences in meteorological factors among the four cities. Pearson correlation and multiple stepwise regressions were used to analyze if the relationship between meteorological factors and COVID-19 infected cases were different among the four cities, and if the relationship can be affected by the season and geographic location.

## 4. Results and Discussion

### 4.1. The Best-Fitting Model of COVID-19 Infected Cases during the Period of Community-Based Epidemic Prevention and Control

Our results found that all *p*-values of the four models were less than 0.1, while R^2^ in the lognormal distribution model were higher than other models (Table 2 and Figure 2). These indicated that the lognormal distribution model had a best-fitting effect on COVID-19 infected cases. Except for the statistical model used in this study, some traditional infectious disease dynamics models, like Susceptible-Exposed-Infected-Removed model (SEIR), were often used in previous studies to fit the change of pandemic infected cases [38,39]. The coefficient of determination (R^2^) of SEIR in these studies fitting the change of COVID-19 confirmed cases ranged from 0.40 to 0.80, which was less than our selected model (R^2^ ≥ 0.86). Pei et al. [40] demonstrated that the SEIR model was not suitable for fitting the change of the infection cases when implemented for high-intensity community-based epidemic prevention and control, due to high-difficulty collection of spatial distribution for the patients and complex modeling process. Compared with SEIR, only one parameter, infected cases, was used. Spatial distribution data were not involved in the lognormal distribution model, which was very simple and easy to use. The lognormal distribution model might have a good application to fit and predict the change of COVID-19 infected cases for the period of community-based epidemic prevention and control.

The fitting effect of the lognormal distribution model to the infected cases was better in Wuhan (R^2^ = 0.89, *p*-value < 0.001) and Beijing (R^2^ = 0.89, *p*-value < 0.001) than in Dalian (R^2^ = 0.86, *p*-value < 0.001) and Urumqi (R^2^ = 0.78, *p*-value < 0.001). This indicated that the fitting accuracy of the lognormal distribution model on COVID-19 infected cases varied among cities. Dalziel et al. [41] and Soebiyanto et al. [42] considered that the fitting accuracy of the pandemic model may be related to virus transmission pattern. Differences in population size and mobility, environmental condition, the quality and quantity of public medical resource, the initial infectious cases and genetic type of virus all affected the variation of COVID-19 spread [23,35,38]. Thus, the transmission pattern would be varied among different cities [35]. This result also can be proven by the difference in model fitting parameters among the four cities. Our result found that the fitting parameters (i.e., *y_0_*, *x_c_*, *w* and *A*) of the lognormal distribution model on COVID-19 infected cases all varied, indicating that the transmission pattern of COVID-19 changed among the four cities (Figure 2).

### 4.2. Influence of Meteorological Factors on COVID-19 Transmission

One-way ANOVA results showed that, except for precipitation (TP), other meteorological factors (T_max_, T_min_, DTR, T_m_, RH and MWS) showed significant difference among the four cities (*p* < 0.05) (Figure 1 and Table 2). Pearson correlation analysis found that TP had no significant relationship with COVID-19 infected cases (*p* > 0.05) (Table 3). This result was consistent with previous studies that rainfall had no direct effect on host susceptibility, virus transmission and survival [6,14,19,42]. Our results showed that the influence of temperature on COVID-19 infected cases varied with their types. T_max_, T_min_ and T_m_ were significantly correlated with COVID-19 infected cases in most situations (*p* < 0.05), while DTR was not related in all cities (*p* > 0.05) (Table 3), suggesting that T_max_, T_min_ and T_m_ had more influence on COVID-19 transmission than DTR. This result was consistent with the temperature–amplitude coupling hypothesis, which implies that virus activity and enzymes necessary to replicate in the host are mainly influenced by the highest, lowest and optimum temperatures, instead of day–night temperature difference [3,6,11,15].

The significant influencing factors of COVID-19 infected cases differed among the four cities. Specifically, COVID-19 infected cases in Wuhan were significantly negatively correlated with T_max_, T_min_ and T_m_ (*p* < 0.05), but not related with other factors (*p* > 0.05). RH had a significant negative correlation with COVID-19 infected cases in Beijing (*p* < 0.05), while MWS showed an opposite pattern (positive; *p* < 0.05). COVID-19 infection cases in Urumqi were significantly affected by T_min_ (negative), T_m_ (negative) and RH (positive) (*p* < 0.05), while not correlated with other factors. COVID-19 infection cases in Dalian were only significantly positively associated with RH (*p* < 0.05) (Table 3). These results indicate that the influences of T_max_, T_min_, T_m_, RH and MWS on COVID-19 transmission differed among the four cities. This was probably due to the superadaptability of the novel coronavirus [10,14,15]. The four cities of this study were located in different climatic regions of China, and the seasons of COVID-19 outbreak also varied among them. According to ecological adaptability theory, the major limiting meteorological factors of viruses vary for different seasons and climatic regions [18,21,24,32]. Similar results were found in previous research, which showed that the contribution of meteorological factors to COVID-19 transmission varied between tropical and temperate regions [11,12,13,14]. For example, high relative humidity mitigated the spread of COVID-19 in tropical regions [14,15], whereas it contributed to the increase of COVID-19 infections in temperate regions [32,43].

The result of multiple stepwise regressions (MSR) differed among the four cities (Table 4). T_min_ was the only last reserved variable of MSR in Wuhan (R^2^ = 0.30, *p* < 0.001). T_m_, RH and MWS were the last influencing factors of COVID-19 infected cases in Beijing (R^2^ = 0.68, *p* < 0.001). T_max_ and RH were the last reserved variables affecting COVID-19 infected cases in Urumqi (R^2^ = 0.27, *p* < 0.05), while T_max_, T_min_ and RH were final reserved variables of MSR in Dalian (R^2^ = 0.75, *p* < 0.001) (Table 4). These results indicate that temperature, humidity and wind speed were most directly related to COVID-19 transmission. This result is consistent with many previous studies [5,11,19,22]. For example, Lowen et al. [28] found that temperature affected the risk of coronavirus infection in humans. Living in a low temperature environment was beneficial for viruses having higher reproductive rate in the upper respiratory tract due to the cooling of mucosa. Novel coronavirus remains active for a long time in the low temperature environment [4,15,16]. A slight rise of temperature will increase spatial distance and exposure risk of the virus due to the increase in the intensity of Brownian motion of atmospheric particulates [27]. Changes in air humidity may injure respiratory epithelial cells and reduce the mucosal ciliary clearance due to the fact that it has an obvious relationship with the moisture conditions of nasal mucosa. The change in air humidity will affect the exposure risk of viruses to humans [11,28]. As wind speed is positively related with the speed and distance of virus transmission in the atmosphere, and the virus is also adsorbed on suspended particles that accumulate near the ground, the increase in wind speed will accelerate the spread of COVID-19 [20,44,45].

The influence of temperature on COVID-19 infected cases varied among cities (Table 4). T_min_ was the most determining factor of COVID-19 transmission in Wuhan, but its contribution on the change of COVID-19 infected cases was less than T_max_ in Beijing, Urumqi and Dalian (Table 4). This difference may be caused by seasonal change. The COVID-19 outbreak in Wuhan occurred in winter (December–March), in contrast to summer in Beijing, Urumqi and Dalian (June–August) (Table 1). COVID-19 transmission in summer may be highly affected by the maximum temperature [46], whereas in winter it might be associated more with the minimum temperature [43]. In summer, the influence of temperature on organisms is mainly implemented by excessive water loss and the inactivation of physiological metabolic enzymes caused by high temperature [6,28,29]. Compared with other living organisms, viruses are more susceptible to high temperatures due to their simple structure and lack of hardened protective tissues. A rise of maximum temperature will induce the denaturation of viral proteins and nucleic acids, inactivating the virus and making it less infectious [28,47,48]. However, in winter, an obvious drop in temperature is beneficial for virus survival and infection. This is probably because the virus is coated with a protective layer of lipid material [29]. In a warm environment, the virus will quickly expire if not within a person or animal body, due to the fact that the protective layer of lipid material needs to remain liquid. During cold winters, a temperature drop will make this layer change from liquid to solid, which subsequently helps protective tissues lengthen the time that the virus can survive in the atmosphere [28,29]. After entering a person’s respiratory tract, the protective tissues will melt and not affect virus proliferation [29]. Thus, COVID-19 depended more on minimum temperature compared with average and maximum temperature in winter. This result was also proven in many current studies. For example, Cervino et al. [49] showed the persistence of coronavirus is favored by a low temperature (4 °C) and is gradually inactivated by the increase of temperature, a finding they reported after summarizing many published articles. Behnood et al. [50] found that COVID-19 had the lowest infection rates at the peak of higher summer temperatures in the United States. Tobías and Molina [51] indicated that minimum temperatures had negative significant relation with COVID-19 confirmed cases in winter in Barcelona. 

The influence of relative humidity on COVID-19 infection cases also differed among the four cities. COVID-19 infected cases had a negative correlation with RH in Beijing and Dalian, while showing a positive relationship in Urumqi. Beijing and Dalian are located in humid areas in eastern China, which are controlled by the oceanic climate and experience abundance air relative humidity (RH > 50%) (Table 5). On the contrary, Urumqi is in an inland desert region with low air relative humidity (RH = 41%) and controlled by the continental climate (Table 5) [52]. This suggests that the influence of relative humidity on COVID-19 infection cases might be associated with geographic location or climate region [6,9,11,13]. The decrease in the relative humidity in humid region might promote COVID-19 transmission, while an opposite pattern occurs in the arid inland region. This conclusion differs from many previous studies showing that a decrease in relative humidity is conducive to coronavirus transmission [53,54,55]. The reason for this difference might be attributed to the relationship between air humidity and the deposition of atmosphere particulates [25,26,27]. Novel coronavirus was mainly transmitted in the atmosphere through droplets exhaled by the virus carrier. These could pass from one person to another through a sneeze, a cough or direct personal contact [49]. Most of it was attached on the surface of a droplet. In humid regions, under the effect of high air humidity, the droplets exhaled by the virus carrier are quickly fused by other droplets into larger drops that quickly sink to the ground when their weight is sufficiently high to overcome the air buoyancy [26,27]. According to the reports from WHO and Cervino et al. [49], the larger droplets (diameter ≥ 5 μm) travel in the air for short distances, generally less than 1 m. The deposition of a large number of massive droplets reduced virus quantity, leading to a negative relationship between air humidity and COVID-19 infected cases [11,38]. However, this explanation does not apply in arid regions such as Urumqi. In drought desert areas, especially in summer, intense evaporation and high temperature pumped more water from the virus and reduced metabolic enzyme activity [56]. Therefore, the increase of air humidity would be conducive to survival and transmission of COVID-19.

Wind speed had a significant positive influence on COVID-19 infected cases in Beijing, but did not affect the other cities. This may be caused by geographic location and season. Beijing is located in the northern part of the North Plains of China, surrounded by the Taihang and Yanshan Mountains, and lies 150 km from the Bohai Sea, which is influenced by both inland and oceanic climate. As the transition from inland to oceanic climates, wind speeds are higher in the period from May and July compared with the other regions studied due to frequently air convection. As Table 1 shows, average wind speed in Beijing was 6.41 m/s, which was significantly higher than the other cities. The advantage of wind speed would be to increase the spread distance and diffusion rate of SARS-CoV-2 coronavirus. Additionally, higher wind speed is also beneficial for the formation of turbulence inside the city, which results in the accumulation of novel coronavirus near the ground, and therefore a higher risk of infection [44,45]. This finding was consistent with previous studies [3,20,57]. For example, Dhruv [3] and Şahin [20] found that wind speed promoted COVID-19 transmission in California and Turkey in spring season, respectively. Yuan et al. [57] indicated that wind speed positively affected the confirmed cases of COVID-19 in Beijing in the spring.

## 5. Conclusions and Comments

COVID-19 infected case records and meteorological data during implementation of community-based epidemic prevention and control were collected in four Chinese cities, in order to test if their relationship was affected by season and geographic location. Compared with other statistical models, the results showed that the lognormal distribution model had the best fit for the changes of COVID-19 infected cases. This indicated that this model can be used to predict the change of the confirmed cases in COVID-19 outbreak cities, especially in cities where community-based epidemic prevention and control in China were implemented. However, the fitting parameters varied among the four cities, indicating that the factors influencing COVID-19 transmission differed. Wind speed, temperature and relative humidity were most directly related to COVID-19 transmission when compared with other meteorological factors, but their influence varied with season and geographic location. In the warm season, the increase in maximum temperature limited COVID-19 transmission in the arid inland region (Urumqi), while high air relative humidity promoted transmission. However, in the more coastal region (Dalian and Beijing), the increases both in maximum temperature and air relative humidity were not conducive to COVID-19 transmission. In contrast, in the cold season, minimum temperature was linked more closely with the spread of COVID-19 than with other meteorological factors. The decrease in the minimum temperature accelerated COVID-19 transmission (Wuhan). Wind speed had a positive relationship with the spread of COVID-19 because it affected the formation of turbulence, which influenced the spread distance and diffusion rate of virus in the atmosphere. In this study, we established regression models to measure the relationship between COVID-19 infected cases and meteorological factors in different seasons and geographical regions (Table 4). R^2^ and *p*-values of all models were greater than 0.27 and less than 0.05, respectively, indicating that our models have certain feasibility in predicting the changes of COVID-19 confirmed cases by using meteorological data. In the future, based on geographic location and season, government and epidemic management agencies can use meteorological forecast data to predict COVID-19 confirmed cases in the short term (such as 15 days) by using our models. This can provide an early warning for reducing the impact of COVID-19 on the lives and health of local residents.

Our results suggest that novel coronavirus is very cunning, can always adapt to adverse circumstances to sustain transmission. The influence of meteorological factors on the propagation and survival of novel coronavirus is complicated and varies with season and geographic location. At present, many studies have proved that meteorological factors, especially temperature, air humidity and wind speed, have significant influence on COVID-19 transmission [7,8,9,10,11,12,13,14,15,16,17,18,19]. However, such studies are mostly drawn on large scales and humid regions [7,8,9,10,11,12,13,14,15]. The design of current pandemic control measures are usually based on the research output of these studies, which may not be applicable for small scales and arid inland cities. In addition, the influence of meteorological factors on COVID-19 transmission changes with season and geographical location. Therefore, the design of community-based COVID-19 prevention and control measures should be conducted for each individual city, and the influence of seasonal changes and difference of geographical location should be accounted for simultaneously. We also found that lower values of both air relative humidity and temperature facilitate the spread of COVID-19 in winter. In the coming months of winter in the northern Hemisphere, increasing indoor temperatures while reducing air relative humidity is an effective way to prevent COVID-19 spread within the community.

## Figures and Tables

**Figure 1 ijerph-18-00484-f001:**
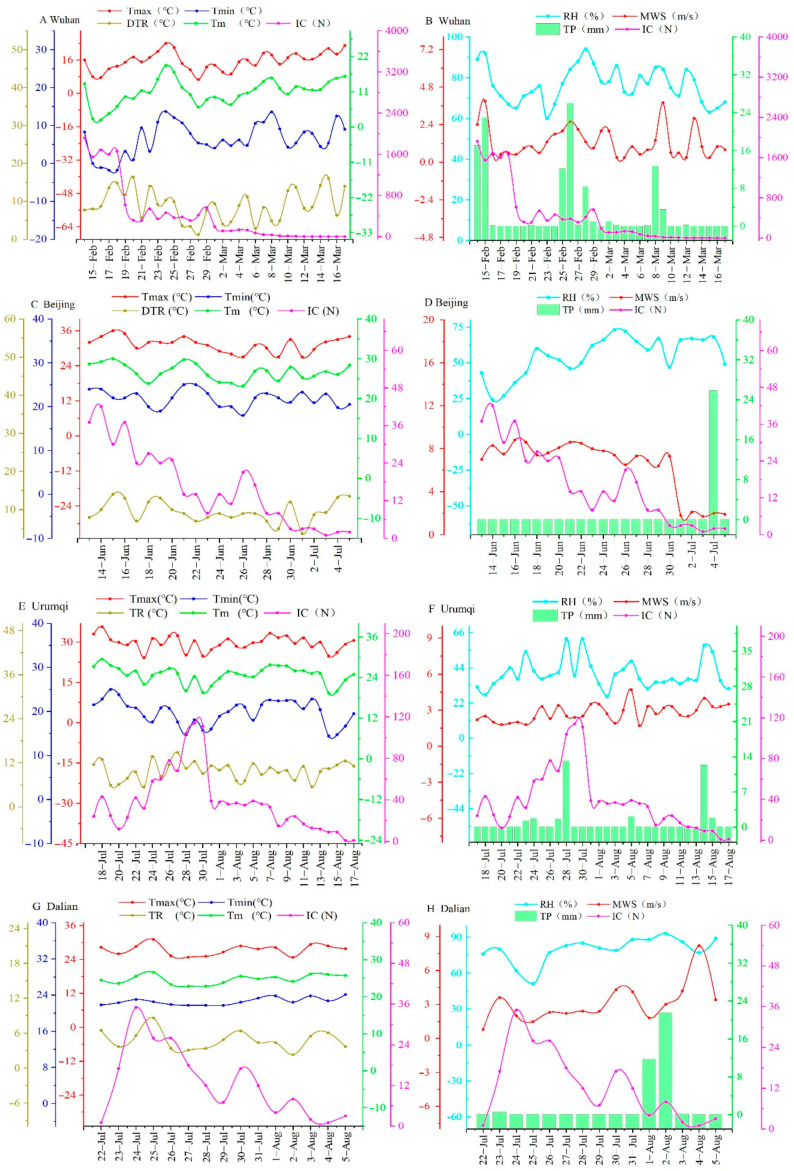
Changes in COVID-19 infected cases and meteorological factors during the period of community-based epidemic prevention and control in four cities. T_max_, T_min_, DTR, T_m_, RH, MWS and TP represent daily maximum temperature, minimum temperature, temperature range, average daily temperature, air relative humidity, average wind speed and precipitation, respectively. IC refers to COVID-19 infected cases.

**Figure 2 ijerph-18-00484-f002:**
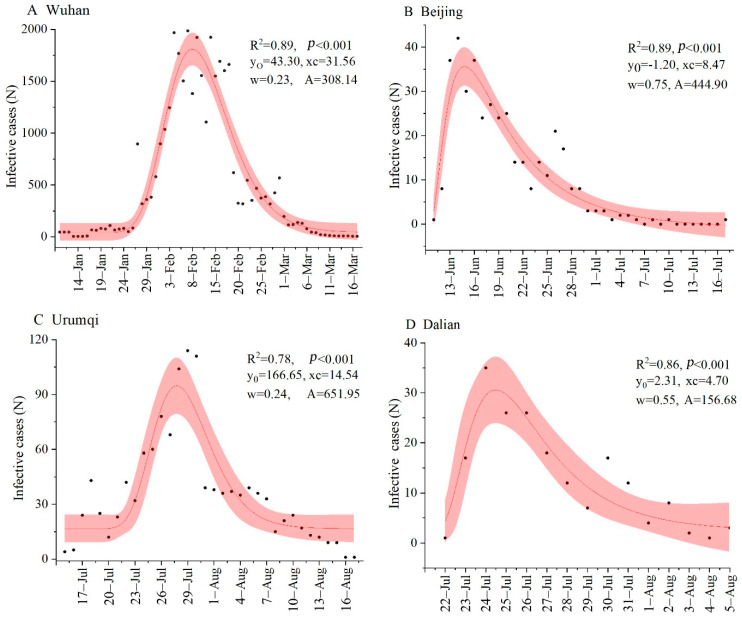
The fitting curve of COVID-19 infected cases using the lognormal distribution model in four Chinese cities (**A**–**D**).

**Table 1 ijerph-18-00484-t001:** Summary of COVID-19 outbreak information for selected cities in China (after December 2019).

Cites	Season	Duration of COVID-19 Outbreaks	Duration of Community Control	Cumulative Number of the Confirmed Cases (N)	Cumulative Number of the Asymptomatic Infected Persons (N)
Wuhan	Winter	8 December 2019–23 March 2020	11 February 2020–27 March 2020	68,100	NA
Harbin	Spring	10 April 2020–21 May 2020	12 April 2020–10 June 2020	66	26
Beijing	Summer	11 June 2020–6 August 2020	13 June 2020–6 July 2020	335	53
Urumqi	Summer	15 July 2020–7 September 2020	13 July 2020–23 August 2020	828	390
Dalian	Summer	22 July 2020–23 August 2020	22 July 2020–20 August 2020	92	97
Qingdao	Autumn	11 October 2020–12 November 2020	12 October 2020–26 October 2020	13	8
Kashgar	Autumn	24 October 2020–at present	24 October 2020–at present	81	423

**Table 2 ijerph-18-00484-t002:** The fitting models of COVID-19 infected cases during the period of community-based epidemic prevention and control. NA indicates no appropriate model to fit the change of COVID-19 infected cases because R^2^ is less than 0.

Models	Equations	Wuhan		Beijing		Urumqi		Dalian	
		R^2^	*p*	R^2^	*p*	R^2^	*p*	R^2^	*p*
Gaussian model	$y=y_{0}+\frac{A}{w\sqrt{\pi /2}}e^{-2\frac{{\left(x-x_{c}\right)}^{2}}{x^{2}}}$	0.88	<0.01	0.74	<0.01	0.78	<0.01	0.48	<0.01
Logistic model	$y=A_{2}+\frac{A_{1}-A_{2}}{1+{\left(\frac{x}{x_{0}}\right)}^{p}}$	NA	NA	0.67	<0.01	NA	NA	NA	NA
Lognormal distribution model	$y=y_{0}+\frac{A}{\sqrt{2\pi }wx}e^{\frac{-{\left(ln\frac{x}{x_{c}}\right)}^{2}}{2w^{2}}}$	0.89	<0.01	0.89	<0.01	0.79	<0.01	0.86	<0.01
Allometric model	$y=ax^{b}$	0.02	<0.01	0.21	<0.01	2.27 × 10^−5^	<0.01	0.11	<0.01

**Table 3 ijerph-18-00484-t003:** Pearson correlation between meteorological factors and COVID-19 infected cases in four cities. Definitions of T_max_, T_min_, DTR, T_m_, RH, MWS and TP are presented in Figure 1; * *p* < 0.05; ** *p* < 0.01; *** *p* < 0.001.

City	T_max_ (°C)	T_min_ (°C)	DTR (°C)	T_m_ (°C)	RH (%)	MWS (m·s^−1^)	TP (mm)
Wuhan (January–March)	−0.42 *	−0.55 **	0.12	−0.53 **	0.15	0.13	0.34
Beijing (June–July)	0.36	0.14	0.24	0.31	−0.67 **	0.65 **	−0.25
Urumqi (July–August)	−0.22	−0.47 **	0.29	−0.39 **	0.42 *	−0.04	0.21
Dalian (July–August)	−0.27	−0.02	0.11	−0.11	−0.65 **	−0.33	0.22

**Table 4 ijerph-18-00484-t004:** The result of multiple stepwise regressions used to analyze the influence of meteorological factors on COVID-19 infected cases (Y) in four cities.

City	Regression Equation	R^2^	*p*-Values
Wuhan (January–March)	Y = 897.06 − 73.24X_Tmin_	0.30	<0.001
Beijing (June–July)	Y = 142.10 − 3.32X_Tm_ − 0.87X_RH_ + 1.65X_MWS_	0.68	<0.001
Urumqi (July–August)	Y = -293.62 + 7.19X_Tmax_ + 2.94X_RH_	0.27	<0.05
Dalian (July–August)	Y = 121.01 − 5.72X_Tmax_ + 6.72X_Tmin_ − 1.30X_RH_	0.75	<0.001

**Table 5 ijerph-18-00484-t005:** Differences in meteorological factors between four cities during the period of community-based epidemic prevention and control, which was tested using one-way ANOVA. Definitions of T_max_, T_min_, DTR, T_m_, RH, MWS and TP are presented in Figure 1. The different lowercase letters after the Mean values indicate a significant difference (*p* < 0.05) in meteorological factors among the four cities, whereas same lowercase letters show no significant differences (*p* > 0.05).

Meteorological Factor	Cites				Statistical Parameter	
	Wuhan (January–March)	Beijing (June–July)	Urumqi (July–August)	Dalian (July–August)	F	*p*-Value
T_max_ (°C)	15.52 ± 4.40c	31.34 ± 2.57a	29.67 ± 2.76a	27.41 ± 1.90b	144.39	<0.001
T_min_ (°C)	6.47 ± 4.30c	21.84 ± 1.87a	19.95 ± 2.84b	22.61 ± 0.83a	173.34	<0.001
DTR (°C)	9.05 ± 4.15c	9.50 ± 2.77b	9.72 ± 2.45a	4.80 ± 1.66d	9.64	<0.001
T_m_ (°C)	10.87 ± 3.93d	26.73 ± 2.01a	24.83 ± 2.59b	24.69 ± 1.24c	193.43	<0.001
RH (%)	76.49 ± 8.82a	54.91 ± 13.87b	40.60 ± 9.34c	79.67 ± 10.85a	78.52	<0.001
MWS (m·s^−1^)	1.29 ± 0.97d	6.41 ± 2.51a	2.73 ± 0.72c	3.08 ± 1.76b	51.89	<0.001
TP (mm)	3.21 ± 6.96a	1.13 ± 5.38a	1.13 ± 3.23a	2.25 ± 6.11a	0.96	0.42

## Data Availability

All data used in this study are issued by China’s National Health Commission and China National Meteorological Information Center. This data can be found here: [http://www.nhc.gov.cn; http://data.cma.cn].

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
