# Peer review of "Influence of Meteorological Factors on the COVID-19 Transmission with Season and Geographic Location"

_ijerph, 2021, doi:10.3390/ijerph18020484_

Round 1

Reviewer 1 Report

Article Title: Influence of meteorological factors on the COVID-19 transmission varied with the season and geographic location

The article brings results of an important investigation relating the influence of meteorological variables and the transmission of COVID-19. The methodology and results typically fit into biometeorological research.

Introduction: The authors update the reader from the initial history of COVID-19 verification in December 2019 in the region of Hubei (city of Wuhan), to its worldwide dissemination, number of cases and effects on the economy and new outbreaks in Europe, with Countries effecting a second closure due to a second wave of the epidemic. They also show that the influence of meteorological variables is not linear, with opposite effects between them in different parts of the world.

The objectives are clear and relevant, since in China there was an efficient control and prevention system for COVID-19, which allows observing the real influence of the seasons and the behavior of the variables analyzed in new registered cases. I emphasize that this study would be (almost) impossible to be replicated in a country like Brazil, where there was no reasonable policy for the containment / dissemination of cases of COVID-19 in the population.

Meteorological data used: They were well described and the justification for the different use of variables between cities was well explained.

NOTE: In topic 3 (Method), the authors cite five methods in full, but show us four among relatives and in Table 2, it is necessary to correct this detail. In addition, the following details need to be corrected in Table 2: I believe that the authors were confused and repeated the p-values ​​of the logistic model and the allometric model in the R2 column, this also happened for Urumqi and only for the logistic model in Dalian.

Results and Discussion: The results are well described, with graphs and explanatory tables relevant to the development of the text. Table 5 shows the regression equations that can even serve as predictive equations, which can be used to output numerical time models and can provide these locations with alerts according to future time perspectives, on a scale of up to fifteen forecast days or more. The expected result is the lack of influence of precipitation, which has already been demonstrated in several biometeorological studies, however, the influence of other variables, according to seasonality and the specific climate of each city, is well demonstrated and discussed.

Conclusion and Comments: It is in accordance with the objective and results achieved and discussed in the article.

Author Response

Response to Reviewer#1

Article Title: Influence of meteorological factors on the COVID-19 transmission varied with the season and geographic location

The article brings results of an important investigation relating the influence of meteorological variables and the transmission of COVID-19. The methodology and results typically fit into biometeorological research.

Introduction: The authors update the reader from the initial history of COVID-19 verification in December 2019 in the region of Hubei (city of Wuhan), to its worldwide dissemination, number of cases and effects on the economy and new outbreaks in Europe, with Countries effecting a second closure due to a second wave of the epidemic. They also show that the influence of meteorological variables is not linear, with opposite effects between them in different parts of the world.

The objectives are clear and relevant, since in China there was an efficient control and prevention system for COVID-19, which allows observing the real influence of the seasons and the behavior of the variables analyzed in new registered cases. I emphasize that this study would be (almost) impossible to be replicated in a country like Brazil, where there was no reasonable policy for the containment / dissemination of cases of COVID-19 in the population.

=> Thank you for reviewing our manuscript and giving us a good comment.

Meteorological data used: They were well described and the justification for the different use of variables between cities was well explained.

=> Thank you for giving us a good comment.

NOTE: In topic 3 (Method), the authors cite five methods in full, but show us four among relatives and in Table 2, it is necessary to correct this detail. In addition, the following details need to be corrected in Table 2: I believe that the authors were confused and repeated the p-values ​​of the logistic model and the allometric model in the R2 column, this also happened for Urumqi and only for the logistic model in Dalian.

=> Thanks for you pointing these errors out. We have changed “five” into “four”. Also, we have changed the R2 in Table 2 in the revised manuscript.

Results and Discussion: The results are well described, with graphs and explanatory tables relevant to the development of the text. Table 5 shows the regression equations that can even serve as predictive equations, which can be used to output numerical time models and can provide these locations with alerts according to future time perspectives, on a scale of up to fifteen forecast days or more. The expected result is the lack of influence of precipitation, which has already been demonstrated in several biometeorological studies, however, the influence of other variables, according to seasonality and the specific climate of each city, is well demonstrated and discussed.

=> Thank you for giving us a good comment.

=> We thank you very much for pointing out that “Table 5 regression equations can even serve as predictive equations”. This is special point we didn't think of and didn't mention in this study. In the revised, we have added some sentences to describe the predictive roles of these models. Please see line 345-353.

Conclusion and Comments: It is in accordance with the objective and results achieved and discussed in the article.

=> Thanks.

Reviewer 2 Report

Dear Authors,

I've really appreciated Your manuscript, it is about a current topic and very interesting.

Manuscript is well conducted and need some minor changes.

In introduction section authors need to better state the aim of the manuscript, maybe they could add a "aim" subsection.

Introduction section is too long and dispersive, please move some data in results section (the table)

In discussion section could be fine to discuss about some similar recent article as:

Cervino, G.; Fiorillo, L.; Surace, G.; Paduano, V.; Fiorillo, M.T.; De Stefano, R.; Laudicella, R.; Baldari, S.; Gaeta, M.; Cicciù, M. SARS-CoV-2 Persistence: Data Summary up to Q2 2020. In Data, 2020; Vol. 5, p 81.

In conclusion section please specify the future perspective of these interesting results.

Author Response

Response to Reviewer#2

I've really appreciated Your manuscript, it is about a current topic and very interesting. Manuscript is well conducted and need some minor changes. In introduction section authors need to better state the aim of the manuscript, maybe they could add a "aim" subsection.

=> Thank you for reviewing our paper and giving us a good comment.

=> We thank you for pointing out that there should be a subsection to describe the aim of study. In the previous manuscript, we wrote aim in the last paragraph of the introduction. However, because it mixed with the epidemic situation in China, it is not beneficial for readers to grasp it. In the revised manuscript, we have reorganized the last paragraph and made it as the subsection to introduce the aim of our study. Please see lines 105-115.

Introduction section is too long and dispersive, please move some data in results section (the table)

=> Thanks. We have moved the Table 1 into Data collection. We did not move Table 1 to the Results section because its content is part of the Data collection section.

In discussion section could be fine to discuss about some similar recent article as:

Cervino, G.; Fiorillo, L.; Surace, G.; Paduano, V.; Fiorillo, M.T.; De Stefano, R.; Laudicella, R.; Baldari, S.; Gaeta, M.; Cicciù, M. SARS-CoV-2 Persistence: Data Summary up to Q2 2020. In Data, 2020; Vol. 5, p 81.

=> Thank you very much for providing us a good article for reference. After reading and learning this article, we are pleasantly surprised to find that many of our results are the same with Cervino et al (2020). We have cited Cervino et al. (2020) in three places of Discussion section to support our results. Please see line 281-283, line 299-301, line 304-308.

In conclusion section please specify the future perspective of these interesting results.

=> Thanks. We have added some sentences in the first paragraph of Conclusion to show the future perspective of these interesting results. Please see line 331-333, line 345-353.

=> We also have re-written the second paragraph of Conclusion to strengthen the future applications of our results. Please see line 354-369.